# Podocytes—The Most Vulnerable Renal Cells in Preeclampsia

**DOI:** 10.3390/ijms21145051

**Published:** 2020-07-17

**Authors:** Ewa Kwiatkowska, Katarzyna Stefańska, Maciej Zieliński, Justyna Sakowska, Martyna Jankowiak, Piotr Trzonkowski, Natalia Marek-Trzonkowska, Sebastian Kwiatkowski

**Affiliations:** 1Clinical Department of Nephrology, Transplantology and Internal Medicine, Pomeranian Medical University, 70-111 Szczecin, Poland; ewakwiat@gmail.com; 2Department of Obstetrics, Medical University of Gdańsk, 80-210 Gdańsk, Poland; 3Department of Medical Immunology, Medical University of Gdańsk, 80-210 Gdańsk, Poland; mzielinski@gumed.edu.pl (M.Z.); justynas@gumed.edu.pl (J.S.); martyna830@gumed.edu.pl (M.J.); ptrzon@gumed.edu.pl (P.T.); 4International Centre for Cancer Vaccine Science Cancer Immunology Group, University of Gdansk, 80-822 Gdańsk, Poland; natalia.marek@gumed.edu.pl; 5Laboratory of Immunoregulation and Cellular Therapies, Department of Family Medicine, Medical University of Gdańsk, 80-210 Gdańsk, Poland; 6Department of Obstetrics and Gynecology, Pomeranian Medical University, 70-111 Szczecin, Poland; kwiatkowskiseba@gmail.com

**Keywords:** preeclampsia, podocytes, VEGF, FSGS, proteinuria

## Abstract

Preeclampsia (PE) is a disorder that affects 3–5% of normal pregnancies. It was believed for a long time that the kidney, similarly to all vessels in the whole system, only sustained endothelial damage. The current knowledge gives rise to a presumption that the main role in the development of proteinuria is played by damage to the podocytes and their slit diaphragm. The podocyte damage mechanism in preeclampsia is connected to free VEGF and nitric oxide (NO) deficiency, and an increased concentration of endothelin-1 and oxidative stress. From national cohort studies, we know that women who had preeclampsia in at least one pregnancy carried five times the risk of developing end-stage renal disease (ESRD) when compared to women with physiological pregnancies. The focal segmental glomerulosclerosis (FSGS) is the dominant histopathological lesion in women with a history of PE. The kidney’s podocytes are not subject to replacement or proliferation. Podocyte depletion exceeding 20% resulted in FSGS, which is a reason for the later development of ESRD. In this review, we present the mechanism of kidney (especially podocytes) injury in preeclampsia. We try to explain how this damage affects further changes in the morphology and function of the kidneys after pregnancy.

## 1. Preeclampsia

According to state-of-the-art research and current knowledge, generalized endothelial damage caused by factors excreted by the placenta into the maternal circulation is the cause of preeclampsia (PE). Angiogenic imbalance leads to epithelial dysfunction. In turn, the imbalance is caused by decreased concentrations of vascular endothelial growth factor (VEGF) and placental growth factor (PlGF), and increased concentrations of soluble fms-like tyrosine kinase-1 (sFlt-1)—a VEGF receptor, and endoglin [1]. The development of preeclampsia is associated with arterial hypertension, proteinuria—usually nephrotic, and decreased glomerular filtration often meeting the criteria for acute kidney injury.

## 2. Glomerular Lesions Secondary to Preeclampsia

The most characteristic histopathological lesion observed in the kidneys of preeclamptic patients is glomerular endotheliosis, which is known to include swollen epithelial cells showing fenestration loss, and fibrin deposits in the subendothelial regions, with both lesions leading to the narrowing or even closing of the glomerular capillaries, and the appearance is that of a “bloodless glomerulus” [2]. Based on the histopathological appearance, it was believed for a long time that the kidney, similarly to all vessels in the whole system, only sustained endothelial damage. Proteinuria was thought to be caused by damage to this part of the filtration barrier. In the glomerulus, the filtration membrane has a unique three-layer structure. Its luminal surface consists of the endothelium, the basement membrane constitutes the inner layer, and the third layer is made of podocytes, with the slit diaphragm sealing the spaces between them. In preeclampsia, two of the filtration membrane components—the endothelium and the podocytes—are damaged, thus leading to proteinuria. Podocytes, with their well-developed contractile apparatus, are capable of regulating the filtration area and the hydraulic resistance of the entire filtration barrier [3]. By contracting their processes, they counter the pressure that inflates the capillaries and thus stabilize the structure of the glomerulus [4]. In a mature glomerulus, podocytes are the only cells participating in the metabolic turnover of the basement membrane, synthesizing its components and producing the proteinases that degrade it [4,5]. Additionally, they produce proteins modulating the properties of the capillary endothelium and are thus regulators of both the expression and function of all the filtration barrier elements [6]. The currently accepted knowledge gives rise to a presumption that the main role in the development of proteinuria is played by damage to the podocytes and their slit diaphragm. 

## 3. Podocytes

As mentioned before, podocytes line the external surface of the glomerular basement membrane. Each podocyte is associated with more than one arteriole, and each arteriole is covered by more than one podocyte. Podocytes are composed of the cellular body, primary processes, and foot processes (or pedicels). The foot processes contain a contractile apparatus including actin, myosin, actinin, talin, vinculin, and vimentin, which opposes the hemodynamic forces of the glomerular capillaries. [7,8] Podocytes’ main task is to participate in glomerular filtration. The glomerular filtrate flows through endothelial fenestrae, the basement membrane, and the slit diaphragms in the spaces between the foot processes. The slit diaphragms are the most important functional elements of the three-layer filtration membrane. They are anchored in the basolateral region of the foot processes. The pedicels are composed of many proteins that form an interacting complex. Damage to one of its elements disorders the function of the slit diaphragms. One of the main proteins of the complex is nephrin, which has an extracellular domain, a transmembrane domain, and an intracellular domain. The extracellular domain forms a network of connections, thus creating the structure for the slit diaphragm, while the intracellular fragment interacts with other proteins, such as CD2AP and CD2-associated protein, podocin, and kinases, passing information from the slit diaphragms on to the podocyte. [9,10] Neph1, a protein similar to nephrin, joins forces with nephrin in building the slit diaphragm structure. Another membrane protein—podocin—binds with the cytoplasmic domain of nephrin and two other proteins—CD2AP and Neph1 [11]. Podocin stabilizes the interacting complex of nephrin, Neph1, and CD2AP. CD2AP is an adaptor protein. It contains five parts, one of which binds with actin. CD2AP is found in all human tissues, although experiments on murine models have shown that normal function only requires its presence in the kidneys [12,13]. Proper interaction of the nephrin-podocin-CD2AP complex is believed to be essential for the flow of fluids, electrolytes, and proteins through the filtration slit to occur [8]. Neph1 also interacts with the protein zonula occludens-1 (ZO-1) [11]. α-actinin that binds actin is another slit diaphragm protein [14]. It is responsible for the contractility of the podocyte processes and the adhesion of pedicels to the basement membrane. Nephrin is believed to bind with α-actinin through podocin. A genetic defect of the proteins forming the slit diaphragm structure—nephrin and Neph1—leads to massive proteinuria. Defects to the intracellular proteins (CD2AP, podocin, and α-actinin-4) cause less pronounced proteinuria, often in later life [15]. Disordered cooperation between the slit diaphragm structure proteins and the contractile element—actin—of the pedicels leads to atrophy of the pedicels, their detachment from the basement membrane, and proteinuria [11]. Podocytes are damaged by detachment from the basement membrane that is associated with the presence of podocytes and their proteins in the urine, or mitotic catastrophe—podocytes may enter the cell cycle, but they rarely undergo mitosis and cannot complete cytokinesis and may undergo apoptosis for variety of reasons. Such damage to the filtration membrane cause proteinuria [6,16,17]. Understandingly, podocytes include other proteins, as well, such as synaptopodin that cooperates with the contractile apparatus, podocalyxin that covers the surface of podocytes giving them negative electric charge, and integrins that attach the pedicels to the basement membrane. Figure 1 shows the scheme of podocytes foot process and slit diaphragm proteins.

## 4. Urinary Excretion of Podocytes and Their Proteins Secondary to Preeclampsia

Many reports are pointing to damage to individual elements of podocytes and slit diaphragms in preeclampsia. As mentioned above, the principal building blocks of the slit diaphragm include nephrin, another job of which is to pass the information on to the pedicels. Ozdemir et al. have found increased concentrations of nephrin in the blood and the urine of patients with severe preeclampsia and intrauterine growth restriction (IUGR). In their study, nephrin levels correlated negatively with fetal weight and age, and positively with creatinine concentration and systolic and diastolic pressure [18]. Wang has found that urine nephrin levels correlate with the severity of proteinuria—in other words, the more severe podocyte and slit diaphragm damage the more pronounced the proteinuria [19]. Jung has noticed that increased urine concentrations of nephrin predate the signs of preeclampsia by an average of nine days [20]. Additionally, other authors have observed that urinary levels of nephrin are higher in PE patients than both in control group women and patients with gestational hypertension [21]. Many reports indicate, as well, the presence of podocytes in the urine, which suggests they have lost attachment to the basement membrane and the filtration membrane has been ruptured. The urinary amounts of podocytes correlate with the severity of preeclampsia [19]. Wang et al. have observed decreased nephrin expression in the excreted podocytes. In their study, they encouraged oxidative stress in the incubated healthy podocyte environment and found that nephrin expression went down. In the same podocytes, they also found downregulated expression of superoxide dismutase—an antioxidant enzyme (CuZn-SOD) [22]. Biopsy specimens from PE patients revealed decreased expression of nephrin compared with the control group [23]. In an experiment in which sFlt and anti-VEGF antibodies were administered intraperitoneally to mice, a similar biopsy specimen appearance was observed. Collino et al. noticed that podocyte incubation with plasma from PE patients did not lead to nephrin depletion. In their study, they subsequently incubated glomerular endothelial cells in PE patient plasma and used the resulting environment to incubate podocytes. They found that nephrin was lost as a result of cleavage of its extracellular domain by proteases and its redistribution. Further on, they established that in response to the PE patient serum, the endothelium produced endothelin 1 (ET-1), which, in addition to being the main cause of activation of the proteases that cleaved nephrin’s extracellular domain, which is the main protein of slit diaphragm [24,25]. The application of recombinant endothelin on cultured podocytes caused the shedding of nephrin from these cells [24]. The same author blocked the activity of VEGF on the glomerular endothelium to find increased endothelin production [25]. Kerlay has analyzed the available clinical studies that assessed urinary podocyte proteins as markers for the development of preeclampsia [26]. In his study, he found that urinary nephrin had the highest sensitivity (0.81) and specificity (0.84) as a marker for the development of preeclampsia [26]. Podocin stabilizes the nephrin-Neph1-CD2AP complex and binds nephrin to α-actinin. This is a key protein in transmitting information from the slit diaphragm to the inside of the podocyte. In his study, Martineau found that preeclamptic patients had higher urinary podocin concentrations than the control group [27]. These concentrations correlated positively with the severity of albuminuria, proteinuria, and arterial pressure, and negatively with gestational age [27]. In his study, Gialni examined podocin-positive extracellular vesicles in the urine and found there were higher levels in PE patients than in the control group. In his paper, he presented the concept that damage to the podocytes is associated with nephrinuria related to nephrin shedding from these cells. According to his claim, this phenomenon causes decreased expression of nephrin in the renal biopsy specimens and in the urine podocytes. Podocin stays bound to the podocytes and its presence in the urine is associated with podocyturia [28]. In his experiment on murine models in which preeclampsia was caused by the administration of a nitric oxide analog, Baijnath identified the mRNA of podocin and nephrin in the urine [29]. As mentioned above, the adaptor protein CD2AP is part of the nephrin-Neph1-Podocin complex. In his study, Henao experimented with podocytes placed in PE patient sera [30]. He found that the distribution of two proteins—podocin and CD2AP—changed. He also noticed that their changed distribution caused an increase in the tension of the contractile apparatus of the podocyte. Additionally, he studied electrical resistance of the podocyte layer and found that, if increased, it suggested low permeability of the filtration membrane, especially for proteins. Podocytes placed in PE pregnant patient serum had lower electrical resistance. The author believed that the changed podocin and CD2AP distribution disordered the entire complex composed of the slit diaphragm and the inside of the foot process containing the contractile apparatus [30]. 

## 5. The Podocyte Damage Mechanism in Preeclampsia

### Free VEGF Deficiency

sFlt-1 that is present in preeclampsia binds with the receptor for VEGF and inhibits its impact on various cells in the system. In the podocytes, especially their foot processes, the expression of all the VEGF-A isoforms is observed. Podocytes are the main sources of VEGF in the glomerulus [31]. A study using the electron microscope established the presence of VEGF in the foot process, the basement membrane, and the luminal surface of the endothelium. It was found that VEGF produced by the podocytes moved in the opposite direction to the glomerular filtrate. VEGF receptors were found both on the podocytes and the endothelial cells. VEGF produced by the podocytes has autocrine and paracrine effects on the endothelium [32,33]. In other studies, mice deprived of podocyte-produced VEGF died upon birth due to renal failure. No normally developed filtration membrane was found in the renal specimens. The problem did not only affect the podocytes, though, as the endothelial cells failed to form the fenestration typical of the glomerulus, as well. Such a phenotype is the responsibility of VEGF produced and secreted by podocytes [24]. The lack of VEGF’s paracrine effect on the endothelium is believed to be responsible for the typical histopathological appearance of the glomerulus in preeclampsia, i.e., glomerular endotheliosis. This experiment proved the influence of VEGF produced by podocytes on the function of podocytes and the endothelium [34]. Podocyte-produced VEGF is bound by heparan sulfate present in the basement membrane, where it is stored and from where it is transported further on. VEGF is known to be necessary for normal podocyte function. It stimulates phosphorylation of nephrin, which prevents podocyte apoptosis [32]. Moreover, VEGF increases interaction between podocin and CD2AP [35]. Administration of anti-VEGF antibodies or sFlt-1 prevents nephrin expression in the podocytes and damages them [33]. In a study on cancer patients treated with anti-VEGF antibodies, podocyturia was observed [36]. Podocytes have a type 1 receptor for VEGF, i.e., VEGFR1, through which VEGF exerts an autocrine effect. No VEGFR-2 receptor was found on their surface. Preeclampsia is known to be accompanied by increased levels of the soluble form of the VEGF receptor, i.e., fms-like tyrosine kinase-1 (sFlt-1), a compound that competes with VEGFR1 for VEGF, which is why podocytes are so exposed to damage in preeclampsia [37]. Experiments have shown that VEGF is necessary to transmit impulses from nephrin (its extracellular domain) to actin—a component of the contractile elements of the podocyte process [35]. Mature mice deprived of VEGF demonstrated damage to all three layers of the filtration membrane [35].

## 6. Endothelial Damage-Related Disorders

### 6.1. Nitric Oxide (NO) Deficiency

Excess amounts of antiangiogenic factors sFlt-1 and endolgin, and a deficiency of angiogenic factors VEGF and PlGF, lead to generalized endothelial damage. This is associated with reduced activity of nitric oxide synthase and decreased levels of this vasodilating factor [38]. In his experiment on murine models, Baijnath administered an analog of l-arginine that inhibited nitric oxide synthesis and caused preeclampsia-like symptoms, including podocyturia, defined as the urinary presence of the mRNA of podocin and nephrin. The histopathological appearance mainly included endothelial damage in the form of swelling, loss of fenestration, and the closing of the vascular walls. This experiment proves the interdependence of the endothelium and the podocytes. The lack of endothelial synthesis of NO causes damage to the podocytes. This mechanism is not yet fully understood. In the same experiment, sildenafil citrate, known to increase cGMP levels (the same result as that of NO), eliminated podocyturia, and prevented histopathological lesions in the glomerulus [29]. It was found that the administration of sildenafil citrate decreased the level of sFlt-1 and increased VEGF synthesis. 

### 6.2. Endothelin-1

Collino’s experiment indicating that the decreased nephrin expression was not caused by the PE patient plasma itself but by endothelin-1, produced by the endothelium under the influence of that plasma, was mentioned above. The application of recombinant endothelin on cultured podocytes caused the shedding of nephrin from the podocytes [24,25]. Preeclamptic patients are observed to have significantly increased plasma concentrations of endothelin-1 [39]. In one study, the administration of blockers of receptors for endothelin-1 before the infusion of sFlt-1 prevented the development of hypertension in murine experiments [40]. 

### 6.3. Oxidative Stress

The abnormal placenta of a PE patient is the source of reactive oxygen species, as well as compounds that damage the endothelium, which itself becomes the source of reactive oxygen species [41]. Wang’s experiment proved that oxidative stress causes damage to podocytes [22]. The podocytes singled out from PE patient urine were shown to demonstrate no expression of nephrin, and none of the superoxide dismutase (SOD) that is normally present on the surface of the foot processes. The author theorized that at this location the job for superoxide dismutase was to protect nephrin (its extracellular domain) against oxidative stress. To prove his thesis, he subjected cultured podocytes to oxidative stress. He achieved the loss of expression of nephrin and superoxide dismutase. He could not prove the direct interdependence of SOD and nephrin but showed that oxidative stress caused nephrin shedding, which had a damaging effect on the filtration membrane [22]. Another author, Zao, studied biopsy specimens from preeclamptic patients to find their decreased expression of nephrin, which proved damage to the podocytes. He also examined a marker of oxidative stress. He found an increased expression of nitrotyrosine and a decreased expression of CuZn-SOD in the biopsy specimens collected from preeclamptic patients when compared with the control group. Nitrotyrosine is a marker of increased oxidative stress and it is formed when a protein molecule is nitrated by peroxynitrite. Superoxide dismutase is the only antioxidant enzyme to dismutate superoxide radicals generated by living cells [42].

## 7. Preeclampsia and the Risk of Developing End-Stage Renal Disease (ESRD)

The effect of past preeclampsia on the women’s health in later life has long been the subject of much debate. Many authors implicate that a history of PE increases the risk of cardiovascular and renal diseases [43,44]. Other studies have shown a higher incidence of microalbuminuria 5 years after preeclampsia [45]. As it was not certain whether that was the result of PE or perhaps co-morbidities that contributed to PE, national cohort studies were carried out. One such study was a Norwegian research paper published in 2008 that was based on the birth registry for 1967–1991, and the ESRD diagnosis registry for 1980 to date. The research showed a higher incidence of ESRD in former preeclamptic patients [43]. Similarly, Swedish national cohort study results published in 2019 confirmed that women who had preeclampsia in at least one pregnancy carried five-times the risk of developing ESRD when compared to women with physiological pregnancies. This correlation was independent of other factors such as co-morbidities, socioeconomic status, or age [46]. Thought needs to be given as to why this happens. In the above work, it was shown that preeclampsia was associated with damage to the podocytes. Podocytes are terminally differentiated cells that do not proliferate or renew. Their damage decreases their number in the glomerulus and leaves a void instead. Podocytes injury is depicted in the following scheme (Figure 2).

## 8. The Mechanism behind Focal Segmental Glomerulosclerosis (FSGS)

There are very few reports on histopathological lesions in the kidneys of patients with a history of PE, which is linked to the ethical contraindications for such studies. However, even those few existing ones suggest that focal segmental glomerulosclerosis (FSGS) is the dominant histopathological lesion [47,48,49,50]. The development of FSGS begins with the loss of podocytes. The kidney of an adult person has approx. 500–600 podocytes per glomerulus, which are not subject to replacement or proliferation. In their study on murine models, Wharram et al. found that podocyte depletion exceeding 20% resulted in FSGS [51]. The development of FSGS because of podocyte depletion has been proven in experimental and clinical studies [52,53]. Podocyte depletion results in a mismatch between the vascular basement membrane area requiring coverage by podocytes and the actual area of the podocytes. The capillaries with the uncovered basement membrane move towards Bowman’s capsule to establish a sort of connection (a cell bridge) with its epithelium Parietal Epithelial Cells (PECs) (lining Bowman’s capsule). Between these bridges, an extracellular matrix gathers that forms fibrous connections with Bowman’s capsule (tuft adhesion). Additionally, after cell bridge formation, PECs de novo express the marker of activation CD44 and start to deposit the Bowman’s-type matrix leading in the tuft adhesion. The CD44-positive PECs are found only in the sclerotic region. [54] Recently are more and more information about the role of PECs in the sclerotic process in secondary FSGS even in PE. [55] Instead of moving to the proximal tubule, the filtrate produced here moves—under Bowman’s capsule epithelial cells—to the peritubular space thus causing tubular atrophy, which then inflicts irreversible damage on the glomerulus. This filtrate has a large protein content as it is not filtrated by the podocyte layer. A further gathering of the extracellular matrix and hyaline substance at the site of the connection with Bowman’s capsule leads to obliteration of the glomerular capillaries. This is where mesangial expansion often occurs [56]. A typical appearance of focal segmental glomerulosclerosis develops. It should be added that proteinuria is not only a sign of filtration membrane damage, as the appearance of proteins in the so-called Bowman’s space causes further damage to the podocytes and Bowman’s capsule epithelial cells and stimulates their apoptosis. In this way, the segmental lesion leads with time to the development of generalized glomerulosclerosis [57]. Additionally, research on children with idiopathic FSGS has shown that proteinuria stimulates the apoptosis of proximal and distal tubular epithelial cells [58]. This provokes the spread of the lesions beyond the glomerulus and across the entire nephron. Moreover, Matsusaka has proven that damage to one podocyte is carried over to the neighboring healthy podocytes, thus causing the domino effect [59]. As a result of the mismatch between the podocyte area and the glomerular basement membrane area, some endothelial cells are not affected by the paracrine effect of VEGF (which is secreted by the podocyte under normal circumstances). This leads to damage to the endothelium—its typical fenestrated phenotype. The changed endothelium may produce compounds that exacerbate podocyte damage [60]. The process, initiated by primary damage to the podocytes secondary to preeclampsia, continues to develop postpartum when the original damaging factors are no longer present. Initially, this causes FSGS-type lesions, and in the long-term leads to the sclerosis of the entire glomerulus and damage to the entire nephron. 

## 9. Conclusions

In preeclampsia, the angiogenic imbalance leads mainly to podocyte damage. The disordered structure of the podocytes leads to their detachment from the basement membrane and urinary excretion (podocyturia). As podocytes do not undergo replacement or proliferation, their numbers are reduced. If the percentage of the damaged podocytes exceeds 20, FSGS-type lesions set in and, being irreversible and progressive, lead with time to the deterioration of renal function. 

A history of preeclampsia is known to be associated with a higher (by five times) probability of developing ESRD, but despite the increased probability, the condition does not affect a large percentage of patients. It is our job as physicians to inform our patients of the need for long-term follow-up monitoring. This should include an albuminuria test, urinalysis, renal function assessment, and arterial pressure measurement on a yearly basis, at least. Patients must be made aware that a past preeclampsia increases the risk of cardiovascular diseases, as well. Apart from reporting for follow-up tests, they could be stimulated to also adopt a healthy lifestyle and avoid any additional risk factors for cardiovascular and renal diseases. The importance of the right diet, exercise, and body weight control, and the need to avoid smoking and alcohol, should be highlighted.

## Figures and Tables

**Figure 1 ijms-21-05051-f001:**
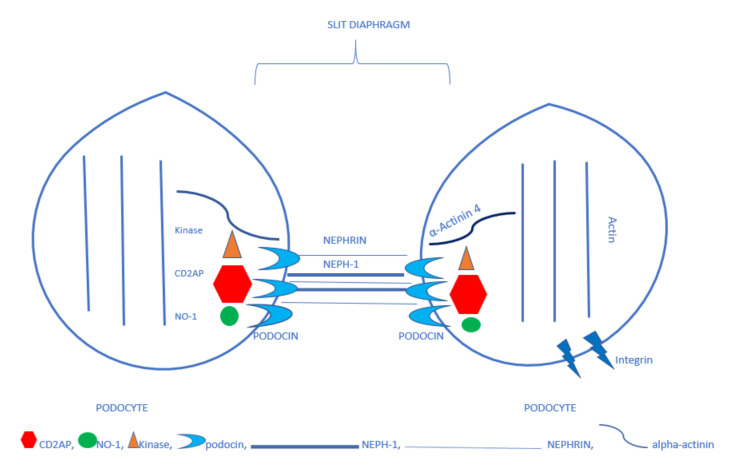
The simplified picture of the construction of a slit diaphragm.

**Figure 2 ijms-21-05051-f002:**
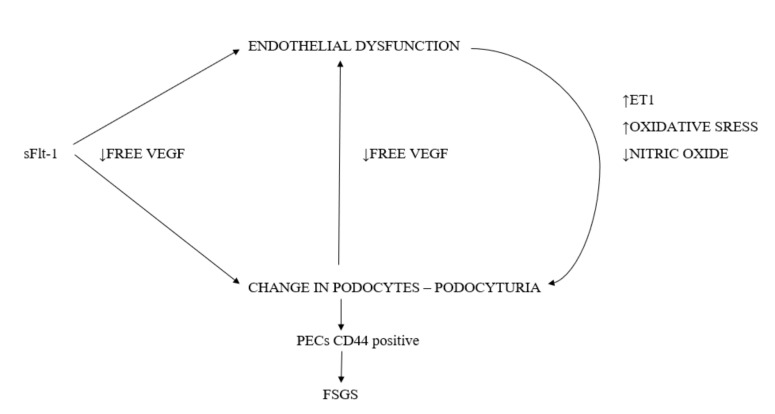
The scheme of podocytes injury in preeclampsia.

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
