# Peer review of "Podocytes—The Most Vulnerable Renal Cells in Preeclampsia"

_ijms, 2020, doi:10.3390/ijms21145051_

Round 1

Reviewer 1 Report

The review by Kwiatkowska et al. is a well-written manuscript that revise an aspect of PE that currently under study but not yet revised. I think that a couple of things need to be fixed before the paper can be accepted.

Major concerns

* When you descried FSGS pathogenesis the Authors describe only marginally the role of PECs in sclerosis development but there are now more and more papers describing how much important is the involvement of these cells in the fibrogenic process (due to an overexpression of CD44), even in PE. One of the proposed mechanism of FSGS is due to podocyte detachment, leading to a podocytes-PECs contact thus leading to PEC activation and sclerosis (for example PMID 25201139). This mechanism should also be added to Figure 2.

*Please check references:

Line 97: The Authors talk about the mitotic catastrophe that was firstly described by the Romagnani group without citing their works.

Line 259-260: The Authors said that podocyte depletion was proven in experimental and clinical studies. The reference they cite id about a clinical study that describes renal biopsies of pregnant women but the Authors of this study never talk about podocytes’ depletion neither of experimental studies in vitro or in animal models.

Minor concerns

* CD2AP should be defined the first time it is named on line 79 instead of line 82;

* Please define IUGR in line 111;

* Figure 1 legend and lines 102-103 are identical, please differentiate;

* The paragraph starting at the end of line 286 is not connected to FSGS mechanisms. I suggest the Authors to move this part to a new paragraph or in the conclusion section.

Author Response

Dear Reviewer 1

Thank you for revision our article entitled “Podocytes – The Most Vulnerable Renal Cells in Preeclampsia”. I made significant changes following your valuable comments.

New sentences are underlined and highlighted in yellow. Deleted sentences are crossed out. Additional articles were cited by the instructions of the reviewers, thus the order of citation was changed.

Major concerns:

When you descried FSGS pathogenesis the Authors describe only marginally the role of PECs in sclerosis development but there are now more and more papers describing how much important is the involvement of these cells in the fibrogenic process (due to an overexpression of CD44), even in PE. One of the proposed mechanism of FSGS is due to podocyte detachment, leading to a podocytes-PECs contact thus leading to PEC activation and sclerosis (for example PMID 25201139). This mechanism should also be added to Figure 2.

In part 8 titled “The Mechanism Behind Focal Segmental Glomerulosclerosis (FSGS)” I wrote new information about the role of PECs in sclerosis development. I cited two additional articles, one proposed by the reviewer and second wrote by Smeets et all. I change Figure 2 adding information about the role of activated PECs in sclerosis formation.

Line 97: The Authors talk about the mitotic catastrophe that was firstly described by the Romagnani group without citing their works.

I cited the work of “Romagnani group” talked about mitotic catastrophe.

Line 259-260: The Authors said that podocyte depletion was proven in experimental and clinical studies. The reference they cite id about a clinical study that describes renal biopsies of pregnant women but the Authors of this study never talk about podocytes’ depletion neither of experimental studies in vitro or animal models.

I cited the articles about podocytes depletion as a cause of FSGS development.

Minor concerns:

CD2AP should be defined the first time it is named on line 79 instead of line 82;

I changed the place of CD2AP explanation from line 82 to 79.

Please define IUGR in line 111;

I defined IUGR on line 111

Figure 1 legend and lines 102-103 are identical, please differentiate;

I change the legend of figure 1

The paragraph starting at the end of line 286 is not connected to FSGS mechanisms. I suggest the Authors to move this part to a new paragraph or in the conclusion section.

The text from part 8 starting at the end of line 286 I moved to the conclusion section.

Sincerely,

Ewa Kwiatkowska

Reviewer 2 Report

The authors review the literature on kidney damage and specifically podocytes in the context of pre-eclampsia (PE). They cover areas of PE and podocyte biology, detecting podocytes and nephrin the urine of PE patients, podocyte damage in PE, endothelial damage-related disorders and the increased risk of developing end-stage renal disease after PE.

This is worthwhile review highlighting the increased risk of podocyte damage and of developing ESRD in women who had preeclampsia in at least one pregnancy, when compared to women with normal pregnancies, and studies on the biological basis behind this.  

The following are some issues to be addressed by the authors:

pg 2 line 65 (section 3: podocytes),

add more references to the beginning of this paragraph

pg 3 line 95:

“podocytes are damaged by apoptosis” is a misleading phrase. Detached podocytes can be viable, and some podocytes may undergo apoptosis for a variety of reasons.  

pg 3 line 98:

“Such damage to the filtration membrane is caused by proteinuria” is a misleading phrase. Proteinuria is a consequence of damage to the filtration membrane.

pg 3, Figure 1:

Some of the labeling in figure 1 is unclear - which proteins are which? Color coding and listing the color scheme in the figure legend would be helpful. What do the lines between the schematic podocytes represent? Are the lines supposed to connect the blue shapes?  

pg 4, line 134: What is meant by “split diaphragm structure”?

pg 5, line 209: no citation listed for “Colin’s experiment” is this the Collino et al. reference?

Author Response

13-07-2020

Dear Reviewer 2

Thank you for revision our article entitled “Podocytes – The Most Vulnerable Renal Cells in Preeclampsia”. I made significant changes following your valuable comments.

New sentences are underlined and highlighted in yellow. Deleted sentences are crossed out. Additional articles were cited by the instructions of the reviewers, thus the order of citation was changed.

Major concerns:

pg 2 line 65 (section 3: podocytes), add more references to the beginning of this paragraph

I added more references to the paragraph 3.

pg 3 line 95: “podocytes are damaged by apoptosis” is a misleading phrase. Detached podocytes can be viable, and some podocytes may undergo apoptosis for a variety of reasons. 

pg 3 line 98: “Such damage to the filtration membrane is caused by proteinuria” is a misleading phrase. Proteinuria is a consequence of damage to the filtration membrane.

pg 4, line 134: What is meant by “split diaphragm structure”?

I change sentence on page 3 – line 95 and 98, and on page 4 line 134

pg 3, Figure 1: Some of the labeling in figure 1 is unclear - which proteins are which? Color coding and listing the color scheme in the figure legend would be helpful. What do the lines between the schematic podocytes represent? Are the lines supposed to connect the blue shapes? 

On figure 1 I added color coding.

pg 5, line 209: no citation listed for “Colin’s experiment” is this the Collino et al. reference?

In the page 5, line 209 I corrected the error from Colin’s experiment to Collino’s experiment.

Sincerely

Ewa Kwiatkowska

Round 2

Reviewer 1 Report

Dear Authors,

I think that the changes you made were fine and improved the manuscript.

I suggest just a couple of minor revisions:

  • line 134 there may be a typing mistake. Did you mean slit diaphragm (split diaphragm)?
  • Figure 1: there are two red lines maybe due to grammatical correction of the software
  • Figure 2: there is an extra line in the last arrow (from PEC to FSGS)

Author Response

14-07-2020

Dear Reviewer

Thank you for the revision of our article entitled “Podocytes – The Most Vulnerable Renal Cells in Preeclampsia”. I made improvements according to your comments. Spelling mistakes and figure’s structures have been corrected and all changes were highlighted in red.

Sincerely,

Maciej Zieliński
